# Three-dimensional imaging of vascular development in the mouse epididymis

**Christelle Damon-Soubeyrand[1†], Antonino Bongiovanni[2†], Areski Chorfa[1†], Chantal Goubely[1], Nelly Pirot[3], Luc Pardanaud[4], Laurence Piboin-Fragner[4], Caroline Vachias[1], Stephanie Bravard[1], Rachel Guiton[1], Jean-Leon Thomas[5,6], Fabrice Saez[1], Ayhan Kocer[1], Meryem Tardivel[2]*, Joël R Drevet[1]*, Joelle Henry-Berger[1]**

[1]Institut « Génétique Reproduction & Développement », UMR CNRS 6293 - Inserm U1103 – Université Clermont Auvergne, Faculté de Médecine, Clermont-Ferrand, France; [2]Université Lille, CNRS, Inserm, CHU Lille, US 41 – UAR 2014-PLBS, Lille, France; [3]Réseau d'Histologie Expérimentale de Montpellier, Institut de Recherche en Cancérologie de Montpellier, Inserm U1194 - Université Montpellier – ICM, Campus Val d'Aurelle, Montpellier, France; [4]Université de Paris, PARCC, INSERM, Paris, France; [5]Department of Neurology, Yale University School of Medicine, New Haven, United States; [6]Institut du Cerveau et de la Moelle, Inserm, Université Pierre et Marie Curie, Paris, France

**\*For correspondence:**
meryem.tardivel@univ-lille.fr
(MT);
joel.drevet@uca.fr (JRD)

†These authors contributed
equally to this work

**Competing interest:** The authors
declare that no competing
interests exist.

**Reviewing Editor:** T Rajendra
Kumar, University of Colorado,
United States

**Abstract** Long considered an accessory tubule of the male reproductive system, the epididymis is proving to be a key determinant of male fertility. In addition to its secretory role in ensuring functional maturation and survival of spermatozoa, the epididymis has a complex immune function. Indeed, it must manage both peripheral tolerance to sperm antigens foreign to the immune system and the protection of spermatozoa as well as the organ itself against pathogens ascending the epididymal tubule. Although our knowledge of the immunobiology of this organ is beginning to accumulate at the molecular and cellular levels, the organization of blood and lymphatic networks of this tissue, important players in the immune response, remains largely unknown. In the present report, we have taken advantage of a VEGFR3:YFP transgenic mouse model. Using high-resolution three-dimensional (3D) imaging and organ clearing coupled with multiplex immunodetections of lymphatic (LYVE1, PDPN, PROX1) and/or blood (PLVAP/Meca32) markers, we provide a simultaneous deep 3D view of the lymphatic and blood epididymal vasculature in the mature adult mouse as well as during postnatal development.

## Editor's evaluation

This fundamental work substantially advances our understanding of vessel development in mouse epididymis. The evidence supporting the conclusions is compelling, with rigorous state-of-the-art microscopy involving high-resolution Three-dimensional imaging. The work will be of broad interest to cell biologists and male reproductive biologists.

## Introduction

Studies regarding the etiology of male infertility have revealed that at least 15% of cases have a clear immunological origin, a share that is most likely underestimated if idiopathic infertilities were to be included (*Dohle et al., 2005*; *Jungwirth et al., 2012*). In the male, they may be the result of chronic inflammatory situations, acute bacterial and/or viral infections, or autoimmune processes along the

genital tract leading to the production of anti-sperm antibodies (for a recent review, see *Shibahara et al., 2021*). These situations are difficult to diagnose and treat, in part because our knowledge of immune/inflammatory responses in male accessory organs is limited. Among the accessory organs, the epididymis plays an essential role in the maturation, storage, and protection of spermatozoa. A particularity of the mammalian epididymis is its highly segmented anatomical organization with three macroscopic territories (caput, corpus, and cauda as depicted in Figure 2), themselves sectorized into segments (10 for the mouse model, see Figure 2), some of which have distinct functional roles (*Turner et al., 2003*; *Johnston et al., 2005*). Unlike the testis, where mammalian evolution has chosen to tightly seal the germline from the immune system—conferring immune privilege to this tissue (*Meinhardt and Hedger, 2011*)—the epididymis tubule faces multiple immune challenges (*Guiton et al., 2013*; *Hedger, 2011*). On the one hand, the epididymis tubule is the testis gatekeeper, preventing ascending pathogens from reaching the immune-privileged seminiferous tubules. To do so, it should be endowed with all the power of a full and efficient immune response toward exogeneous antigens. On the other hand, the epididymis must maintain self-tolerance toward spermatic antigens that are unique to this cell lineage and appear at puberty long after the establishment of the self-immune repertoire (*Goodnow, 1996*). Understanding how this dual action of peripheral tolerance toward sperm antigens (*Mueller, 2010*) versus efficient immune survey toward non-self-antigens is performed is of particular interest for infertility issues, but also more largely in other settings where this situation occurs. This has prompted recent research aiming at identifying immune cells and molecules that could participate in the finely orchestrated epididymal immune physiology. This research has led to the characterization of a dense network of peritubular antigen-presenting cells (*Da Silva et al., 2011*), interstitial and intra-epithelial lymphocytes of distinct sub-families (*Voisin et al., 2018*), as well as of immunosuppressive players such as transforming growth factor beta (*Pierucci-Alves et al., 2018*; *Voisin et al., 2020*) and indoleamine/tryptophane dioxygenase activity (*Britan et al., 2006*; *Jrad-Lamine et al., 2011*; *Jrad-Lamine et al., 2013*).

Because the immune picture of the mammalian epididymis would not be complete without a clear view of the blood and lymphatic circulating networks—which are crucial in managing immune responses—we choose to explore these in the mouse. Reports describing mammalian epididymal blood and lymphatic vessels are rare and rather old (*Pérez-Clavier et al., 1982*; *Abe et al., 1984*; *McDonald and Scothorne, 1988*; *Itoh et al., 1998*). The technology used was not very efficient and consisted mainly of ink or contrast agent injection. It concerned only the peripheral epididymal lymphatic network with the organization of the collectors connecting the PP. More recently, *Hirai et al., 2010*, performed the first histological study of mouse epididymal blood and lymphatic vasculature using the markers CD31/PECAM1 and LYVE1, respectively. Although PECAM1 is used to visualize blood vessels, it is not an exclusive blood marker because it is also found expressed at button-like junctions of endothelial cells in the initial lymphatics (*Privratsky and Newman, 2014*). Furthermore, to characterize lymphatics, the use of LYVE1 alone limits the power of the analysis because this marker recognizes only a subpopulation of all lymphatics (*Kato et al., 2006*). In addition, conventional two-dimensional (2D) light microscopy combined with a horseradish peroxidase (HRP) reporter on paraffin-embedded tissue sections did not allow for a high level of accuracy in providing a holistic view of a structured network.

Characterization of the lymphatic vasculature in mammals is tricky because on the one hand, it shares common embryonic origins with the blood vessels (*Baluk and McDonald, 2008*; *Srinivasan et al., 2007*) and on the second hand, it concerns various different structures. There are blunted-end highly permeable initial lymphatics connected to pre-collector lymphatics and then to collector lymphatics linking to a lymph node (LN), each with specific characteristics (for a very recent review, see: *Lampejo et al., 2023*). A step forward was made with the identification of lymphangiogenic factors (vascular endothelial growth factors VEGF-C and VEGF-D) and their receptor (VEGFR3), which are now considered the major markers of lymphatics (*Kaipainen et al., 1995*; *Mäkinen et al., 2001*). To ensure confidence in the identification of lymphatic structures, one should use a combination of markers, including VEGFR3, the lymphatic vessel endothelial hyaluronan receptor 1 (LYVE1), the prospero-related homeobox 1 (PROX1) transcription factor, and podoplanin (PDPN) (*Banerji et al., 1999*; *Breiteneder-Geleff et al., 1999*; *Hamrah et al., 2003*; *Kivelä et al., 2016*; *Mouta Carreira et al., 2001*; *Partanen et al., 2000*; *Wigle and Oliver, 1999*). We have followed this rationale in the present study, using three lymphatic markers together with plasmalemma vesicle-associated protein

(PLVAP/MECA32) to specifically identify fenestrated blood vessels (*Stan et al., 2004*). In addition, we have taken advantage of a transgenic mouse model that expresses the fluorescent reporter YFP under the control of the VEGFR3 promoter (*Calvo et al., 2011*). Using the power of three-dimensional (3D) reconstruction of high-resolution imaging with light-sheet microscopy after epididymis 3DISCO clearing (*Belle et al., 2014*; *Ertürk et al., 2012*), we present here an in-depth analysis of both the blood and lymphatic networks of the epididymis in the adult mouse as well as during postnatal onto-genesis. We hope the data we bring will provide a better view of the systemic compartment that irrigates and drains this segmented organ, a prerequisite for a more precise understanding of its pathophysiology.

## Results

### An abundant lymphatic network drains the epididymis

To gain insight into the lymphatic network of the mouse epididymis, we used the VEGFR3:YFP mouse model. *Figure 1* shows representative in toto views of the lymphatic vascularization of the testis and epididymis of an adult (*Figure 1A*) and 10 days postnatal (DPN) mouse (*Figure 1B, C*). There is intense vascularization at the epididymis–testis interface regardless of the level of the epididymis (*caput*, *corpus*, or *cauda*). *Figure 1B* shows that the lymphatic drainage joins the main testicular lymphatic trunk at the *pampiniform plexus* (PP) and follows it to a highly fluorescent structure that corresponds to the most proximal LN. Inset 1C and *Figure 1D* show the connections of the collectors emanating from the *caput* (upper epididymal lymphatic collector [u-ELC] in *Figure 1C, D*) and the *corpus* (median epididymal lymphatic collector [m-ELC] in *Figure 1C*). A higher magnification micrograph (*Figure 1D ①*) shows the particularly prominent lymphatic vasculature of the efferent duct and its interconnection with that of the epididymal *caput*. It also points out the departure of the u-ELC to the PP surrounding the upper epididymal artery. It can be noted that the lymphatic collectors coincide well with the segmentation of the organ. A *corpus* lymphatic collector (*Co*-LC) follows the body of the epididymis on the side adjacent to the testis connecting the *caput* with the *cauda* (*Figure 1E*). The inset within *Figure 1E* presents a higher magnification of the *Co*-LC connections. This collector also connects to the main testicular lymphatic collector at the *corpus* level (*Figure 1E ②*) as well as at the *cauda* level (*Figure 1F ③*). Also noticeable in the enlargements provided in *Figure 1E* and in *Figure 1F* is the presence of fluorescence under the capsule, faint in the *caput* (*Figure 1E*) but clearly more visible in the most distal segment (S10) of the *cauda* epididymis (*Figure 1F and G*). Finally, *Figure 1G* presents a global complex organization of the *cauda* lymphatic collectors. Overall, our observations revealed a very dense lymphatic network in the mouse epididymis.

### Complexity of the blood and lymphatic vascularization of the epididymis

Because fenestrated blood vessels also express VEGFR3, a marker specific for these vessels (PLVAP/Meca32) was used along with three additional markers for lymphatics, specifically LYVE1, PROX1, and PDPN. As the 3DISCO organ clearing procedures resulted in a loss of the endogenous transgene (VEGFR3:YFP) fluorescence, anti-GFP was used to strengthen the signal. The concordance of the labeling obtained with the anti-GFP and an anti-VEGFR3 antibodies was controlled (see *Figure 2— figure supplement 1A*). Similarly, *Figure 2—figure supplement 1B and C* show positive controls of the multiplex labeling as well as controls with the secondary antibody, respectively. *Figure 2A* and *Figure 2—videos 1–3* show the immunoreactive vasculature after 3DISCO clearing of an adult mouse epididymis obtained after light-sheet imaging and IMARIS 3D reconstruction. The *caput* and *cauda* territories show intense PLVAP reactivity revealing a dense blood network. The *corpus* of the epididymis is also well vascularized, as can be seen in *Figure 2—figure supplement 2A* and *Figure 2— video 2*. The video clearly shows that the lymphatic network is asymmetrical as it is very dense on the epididymal side next to the testis. In agreement with previous reports (*Abe et al., 1984*; *Suzuki, 1982*), the initial segment (IS) of the *caput* shows the highest reactivity (see *Figures 2B and 3A* as well as *Figure 2—figure supplement 2A and B*), with each IS tubule strongly reactive at their periphery (*Figure 2—figure supplement 3A and B*). Outside the IS, PLVAP reactivity is mainly found in the inter-tubular space as well as along the septum separating each epididymal segment. This is particularly true for the IS/S2–S3 and the S9/S10 septum junctions.

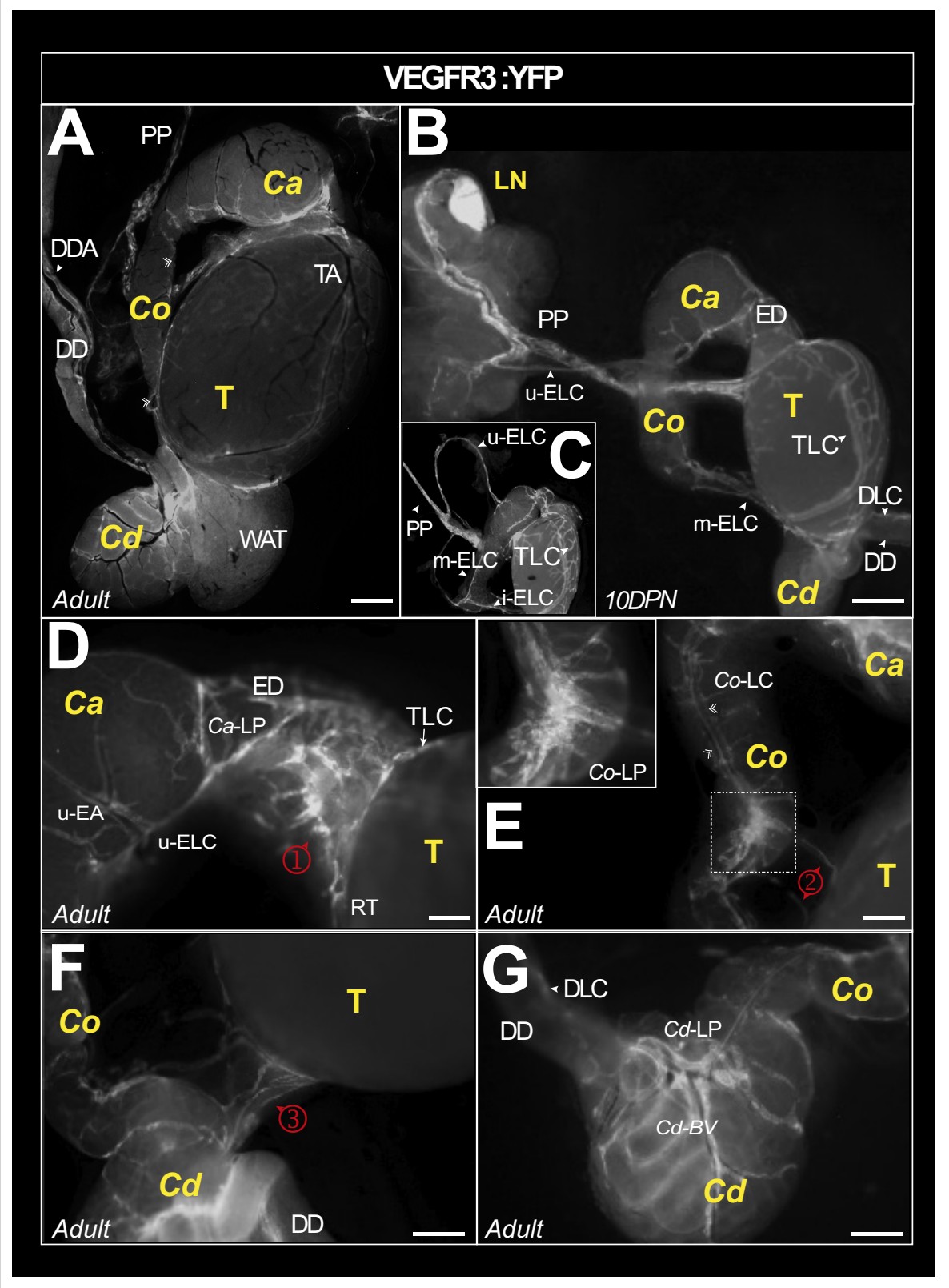

**Figure 1.** Macroscopic view of the lymphatic vasculature of the epididymis and testis in the VEGFR3:YFP model. Representative image of the lymphatic vasculature of the adult epididymis and testis observed with a Leica binocular loupe (**A**). VEGFR3:YFPpos lymphatics are visible in the caput, corpus, and cauda regions. Lymphatics responsible for epididymal and testicular drainage follow the pampiniform plexus (PP) before reaching a lymph node (**B**). The superior and lateral epididymal lymphatic collectors (upperELC and medianELC, respectively) drain lymph from the caput and corpus, respectively,

*Figure 1 continued on next page*

*Figure 1 continued*

before joining the main testicular lymphatic collector at the PP. The latter is connected to a main collector that surrounds the testis (arrowhead) and branches into a rich network (**A–C**). There are numerous lymphatic connections between the lymphatics of the epididymis and the main lymphatic collector of the testis, notably through a lymphatic network at the level of the efferent ducts (φ in D), corpus (κ in E), and cauda (λ in F). Through the capsule covering the epididymal duct, there is also fluorescence that outlines the tubules, weakly in the caput (**D**) and more intense in the cauda (**F and G**). The scale bar corresponds to 1 mm. AT = adipose tissue; Ca = caput; Cd = cauda; Co = corpus; DD = deferent duct; DDA = deferent duct artery; DLC = deferent lymphatic collector; DPN = days postnatal; ED = efferent duct; LELC = lower epididymal lymphatic collector; LN; lymph node; P=pampiniform plexus; SEA = superior epididymal artery; Cd-BV = caudal blood vessel; SELC = superior epididymal lymphatic collector; T = testis; TA = testis artery; TLC = testicular lymphatic collector. ①=caput–testis lymphatic connection; ②=corpus–testis lymphatic connection; ③=cauda–testis lymphatic connections between testis and epididymis.

With the four lymphatic markers (VEGFR3:YFP, LYVE1, PROX1, and PDPN), there is an intense and complex network (*Figure 2B*). This appears very similar in its organization to that of the blood vessels, especially in the IS (*Figure 2B*). However, outside the IS, lymphatic reactivity appears more extensive than that of the blood vessel marker (see *Figure 2—figure supplement 3B*). As was the case for the blood vessel marker and as expected because fenestrated vessels express VEGFR3 (*Partanen et al., 2000*), lymphatic marker reactivity is stronger in the IS (*Figure 2B*). As seen in *Figure 1*, large external lymphatic collectors follow the septa, particularly at the IS/S2-S3 and S9/S10 boundaries. These collectors can be seen to dip inside the organ and irrigate the intertubular compartments, especially in the *cauda* region (see *Figure 2—video 3*). The LYVE1, PROX1, and VEGFR3:YFP profiles are very similar for the *caput* and *cauda* (*Figure 2B*). However, when viewed at higher magnification (*Figure 2—figure supplement 3A*), they do not merge totally in either the *caput* or the *cauda* region, especially for the small lymphatic vessels separating the epididymal segments. The PDPN profile appears different from those of LYVE1, PROX1, and VEGFR3:YFP (*Figure 2B*). The PDPN reactivity is more homogeneous and is concentrated at the peritubular level (*Figure 2B* and *Figure 2—figure supplement 3A*). *Figure 2B* and *Figure 2—figure supplement 3A* also show that the lymphatic marker (PROX1) gives very similar results to LYVE1 in the *caput* and *cauda* regions. However, the pattern of PROX1 is slightly distinct from that of VEGFR3:YFP and very different from that of PDPN.

We performed a surface assessment to better evaluate the density of the blood and lymphatic networks in the most proximal and distal epididymal segments (IS and S10) compared with that observed in the rest of the *caput* or *cauda*. To measure only the specific density of the lymphatic network, voxels with signal in the 555 nm channel (PLVAP/MECA32) were set to 0 in all other channels. The densities presented in *Figure 3A and B* are the results of the vessel volume normalized by the total volume of the region of interest (IS, *caput*, S10, or *cauda*). As expected, PLVAP/MECA32$^{pos}$ blood vessel density in the IS was approximately 3.5 times greater than in the rest of the *caput* (p<0.001). Similarly, the VEGFR3$^{pos}$ lymphatic density was twice that of the rest of the organ (*Figure 3B*). A result of the same magnitude was obtained for LYVE1 and PROX1 (*Figure 3B*). Because of the expression profile of PDPN, surface rendering could not be used to calculate its density. In the *cauda*, there was no significant difference in the density of blood and lymphatic vessels. Comparison of vascular density between the *caput* (IS excluded) and the *cauda* showed a slight decrease in the *cauda* for both blood and lymphatic vessels (*Figure 2—figure supplement 2B*).

## Dynamic development of epididymal vasculature related to organ maturation

To better understand the postnatal development of both networks, we followed their emergence during postnatal epididymal development from 10 to 40 DPN, when the organ is functionally mature (n=5). We present in *Figure 4A* (*caput*) and *Figure 4—figure supplement 1A* (*cauda*) representative high-resolution images of the blood and lymphatic systems using three of the five markers presented above (PLVAP/MECA32, VEGFR3:YFP/GFP, and LYVE1) at 10, 20, 30, and 40 DPN. We used IMARIS software to estimate the IS volume relative to the *caput* volume (*Figure 4B*) and the S10 volume relative to the *cauda* volume (*Figure 4—figure supplement 1B*) at each time point.

IS progresses four times faster than the *caput* between 10 DPN and adulthood (A), representing less than 10% of the *caput* at 10 DPN whereas it represents 40% of the adult *caput* (*Figure 4A and B*). Following the stronger postnatal development of the IS, we observed that the *caput* vascularization is rather homogeneous at 10 DPN, whereas from 20 DPN onward, the blood vessels seem to

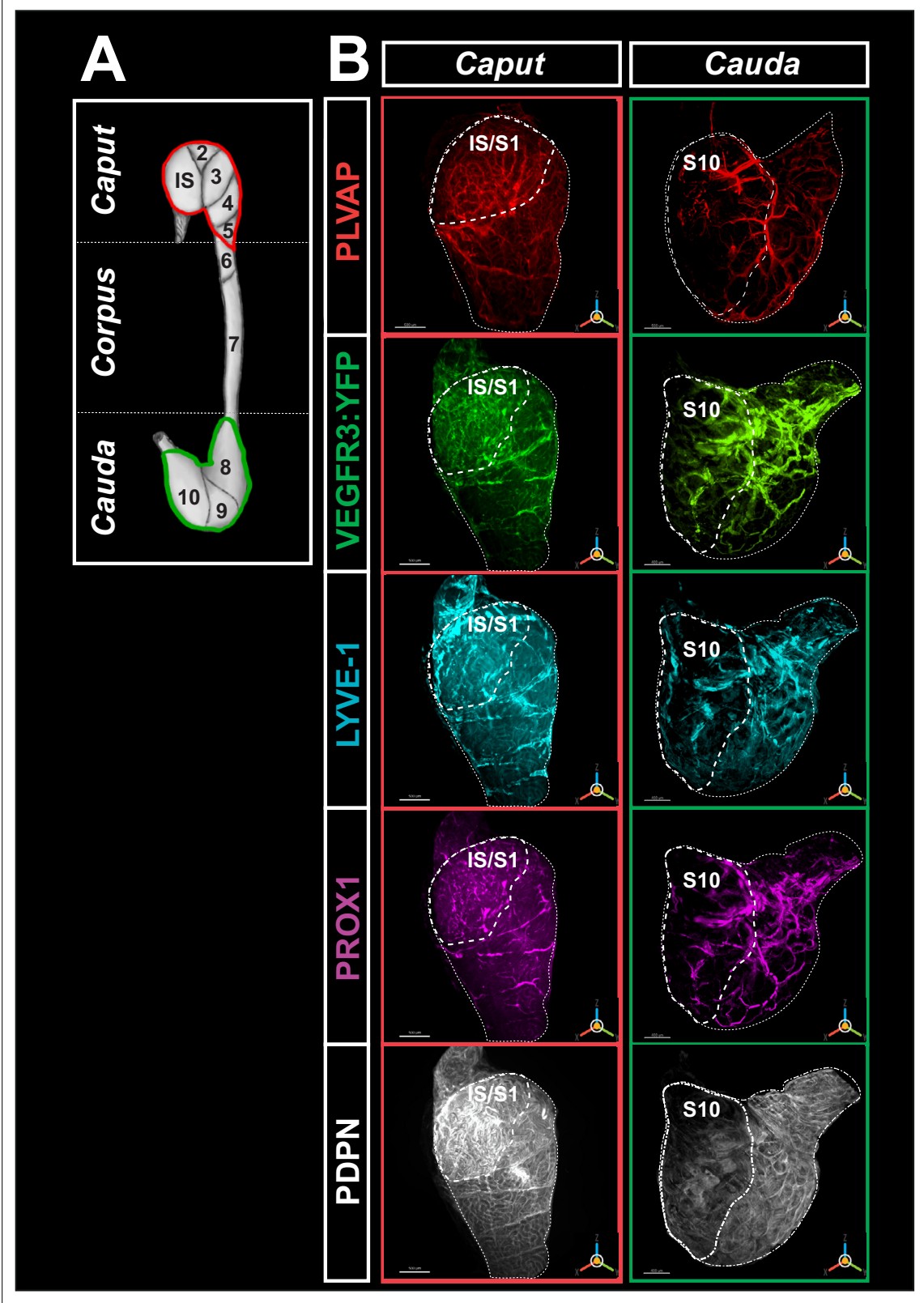

**Figure 2.** High-resolution three-dimensional (3D) imaging of the blood and lymphatic vasculature of the mouse epididymis after organ clearing. (**A**) shows a schematic representation of the caput (red) and cauda (green) regionalization and segmentation commonly used to describe the mouse epididymis. (**B**) Representative multiplex immunostaining of the caput (left) and cauda (right) of the epididymis using five markers recognizing blood and lymphatic vessels is shown. An anti-MECA32 antibody (red) reveals PLVAP^pos blood vessels, particularly fenestrated vessels. An anti-GFP

*Figure 2 continued on next page*

*Figure 2 continued*

antibody revealing the VEGFR3:YFP transgene (green) and an anti-LYVE1 antibody (cyan). Anti-LYVE1 (cyan), anti-PROX1 (magenta), and anti-PDPN (white) antibodies were used as additional lymphatic markers. After organ clearance, high-resolution 3D imaging was performed with a light-sheet ultramicroscope (LaVision BioTec). The two extreme segments, IS (initial segment or S1) and S10, are delineated by a large dashed line, whereas the caput (left) and cauda (right) regions are delineated by a small dashed line. The scale bar is 500 μm. The PLVAP image corresponds to the contro-lateral epididymis used for the four lymphatic markers since it is not possible to use all five markers on the same organ.

The online version of this article includes the following video and figure supplement(s) for figure 2:

**Figure supplement 1.** Positive and negative immunohistochemical controls.

**Figure supplement 2.** Three-dimensional (3D) view of blood and lymphatic vascularization of whole epididymis after 3DISCO clarification.

**Figure supplement 3.** Cross-sectional views of the clarified caput and cauda epididymides.

**Figure 2—video 1.** Three-dimensional (3D) imaging of multiplex labeling of blood and lymphatic networks in the adult mouse *caput* epididymis. https://elifesciences.org/articles/82748/figures#fig2video1

**Figure 2—video 2.** Three-dimensional (3D) imaging of multiplex labeling of blood and lymphatic networks in the adult mouse *corpus* epididymis. https://elifesciences.org/articles/82748/figures#fig2video2

**Figure 2—video 3.** Three-dimensional (3D) imaging of multiplex labeling of blood and lymphatic networks in the adult mouse *cauda* epididymis. https://elifesciences.org/articles/82748/figures#fig2video3

concentrate in the IS segment. This is also well illustrated in the densitometric quantification shown in *Figure 4C*. This figure also shows that the density of PLVAP$^{pos}$ blood vessels in the IS segment is rather stable during postnatal development despite the increase in the segment volume. In contrast, there is a strong decrease between 10 and 20 DPN in the rest of the *caput*. In the *cauda* region (*Figure 4—figure supplement 1A–C*), PLVAP$^{pos}$ blood vessel expansion follows organ growth without a specific segmental pattern. However, as was the case in the *caput*, between 10 and 20 DPN, there is a decrease in PLVAP$^{pos}$ blood vessel density in the caudal region.

Looking at the two lymphatic markers (VEGFR3:YFP and LYVE1), we observed very similar kinetics of postnatal lymphatic development in the *caput* (*Figure 4A*) and *cauda* (*Figure 4—figure supplement 1A*). There is a progressive increase in VEGFR3:YFP/LYVE1 lymphatic vessels that follows postnatal organ growth in a fairly linear fashion at least between 10 and 40 DPN (*Figure 4D* and *Figure 4—figure supplement 1D*). Beyond 40 DPN, there appears to be a greater expansion of lymphatics in the *caput* and *cauda*, in the IS and S10, respectively, compared with the rest of the organ (*Figure 4D* and *Figure 4—figure supplement 1D*). This is particularly true in the IS because the size of this territory is increased approximately four times between the 10 and 40 DPN stages.

Because blood and lymphatic vascular density data suggest neovascularization during postnatal epididymal development, we performed localization and quantification of angiogenic (VEGF-A) and lymphangiogenic (VEGF-C and VEGF-D) factors, whose receptors are VEGFR2 (also known as KDR and FLK1) and VEGFR3, respectively (*Figures 5 and 6*). Quantification of the VEGFR ligands (VEGF-A, VEGF-C, and VEGF-D) was performed by western blotting as shown in *Figure 5A*. For the angiogenic ligand (VEGF-A), there was a decrease between 10 and 20 DPN followed by a plateau, and then an increase at 40 DPN; by adulthood, VEGF-A had returned to the initial level (*Figure 5B*). For the lymphangiogenic ligands (VEGF-C and VEGF-D), we observed a linear increase from 30 DPN for both ligands, consistent with the accumulation of their corresponding receptor VEGFR3 (*Figure 5B*). An analysis of covariance of the different ligand isoforms of VEGF (see *Figure 5—figure supplement 1*) with respect to the VEGFR3 receptor (see *Figure 5C*) shows strong statistical significance for the active ligand isoforms (i.e., the 21 kDa form for VEGF-C and the 24 kDa form for VEGF-D). In addition, the tissue localization of VEGFR2 is very similar to that of VEGFR3 (*Figure 6A*), being present mainly in a peritubular position at the level of the IS whereas in the more distal part of the organ it seems to be restricted to some interstitial vessels (used with permission from Dr. A. Medvinsky; VEGFR2:GFP mouse strain; *Xu et al., 2010*). In accordance with previous work (*Korpelainen et al., 1998*), the tissue localization of its angiogenic ligand (VEGF-A) seems to correspond correctly (*Figure 6A*), supporting the intense blood vascularization observed in this very proximal segment of the epididymis. Both of the lymphangiogenic ligands—VEGF-C and VEGF-D—show a similar tissue localization, being weakly present in the IS and increasingly present when moving toward the tail of the epididymis, mainly in an interstitial position (*Figure 6B*). A co-localization (Pearson coefficient = 0.684 ± 0.038) and

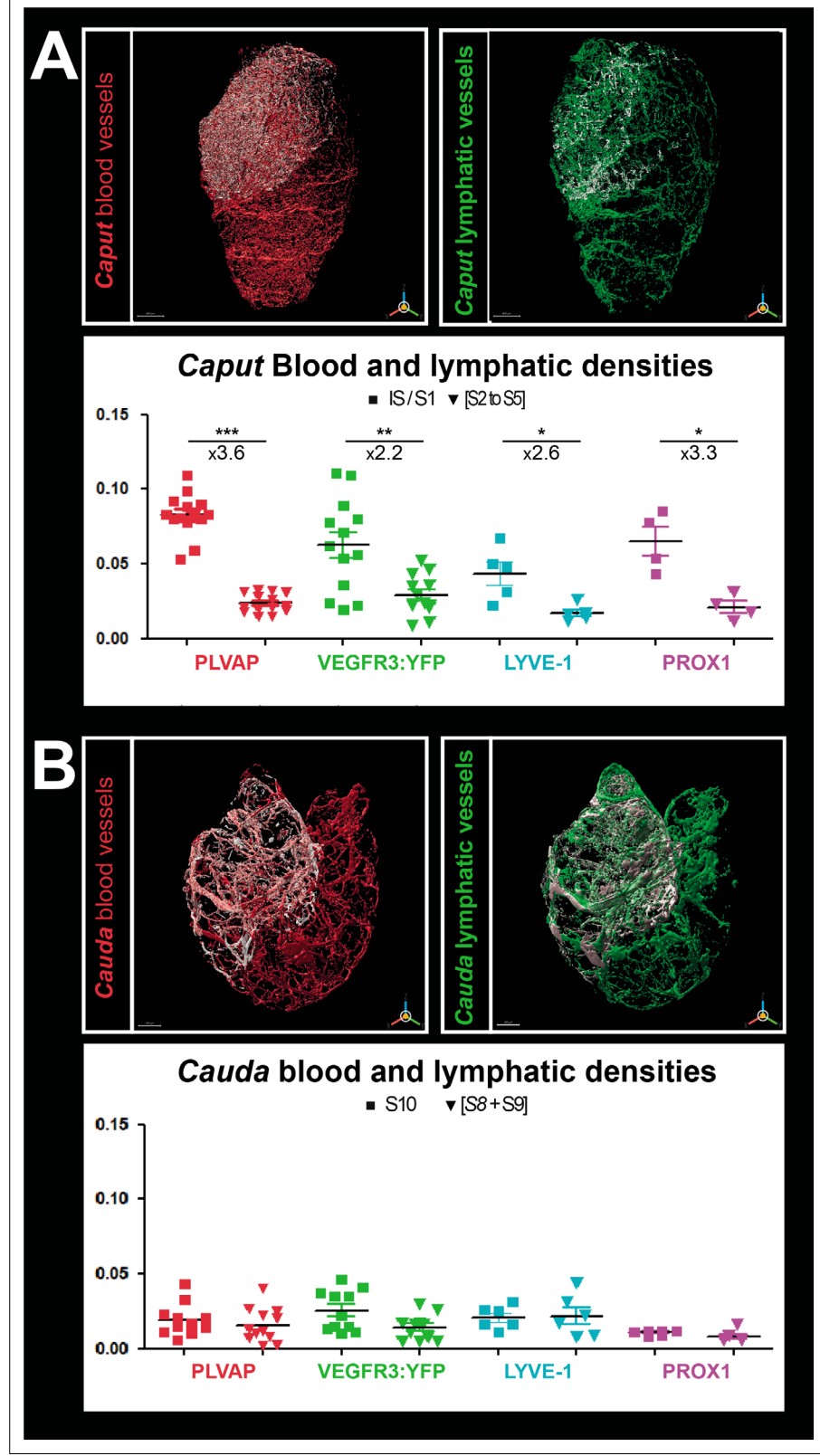

**Figure 3.** Densitometry of the blood and lymphatic vasculature of the mouse epididymis. Panel A shows the surface rendering performed with IMARIS software of blood vessels (red, left) and lymphatic vessels (green, right) in the caput. The surface rendering of vessels in S1 is in white in both cases. Vessel densities shown in the graphs below correspond to the ratio of the volume occupied by vessels in the S1 or caput (minus S1) normalized to the

*Figure 3 continued on next page*

*Figure 3 continued*

total volume of the S1 or caput (minus S1), respectively. Panel B shows the surface rendering of blood (red, left) and lymphatic (green, right) vessels in the cauda. Blood or lymphatic vessel densities are measured as described in A for the caput. The Mann–Whitney test was used to determine statistical significance (***p<0.0001, **p<0.001, *p<0.01, NS = not significant).

co-occurrence analysis (Mander's coefficient VEGF-C=0.934 ± 0.028 and VEGF-D=0.569 ± 0.031) supports the hypothesis of heterodimerization of the two ligands.

Lymphatics expressing all four markers (VEGFR3:YFP, LYVE1, PROX1, and PDPN) are particularly visible at the septa (*Figure 7A*). We show in *Figure 7B* (upper panels) a dynamic 3D view of the *cauda* epididymis lymphangiogenesis as well as our interpretation of what is happening, presented in the lower panels of *Figure 7B*. LYVE1[pos] cubblestone cells are observed at 10 DPN while very few VEGFR3-YFP[pos] cells are visible. Later, cubblestone cells appear to aggregate to form tubular structures. Concomitantly, we record an increase in VEGFR3-YFP[pos] structures associated with a decrease in PLVAP[pos] vessels. This kinetics of events is consistent with active lympho-vasculogenesis during postnatal epididymal development. The schematic shown in *Figure 8* presents the detailed organization of the epididymis lymphatics with respect to the tubules, septa, and orientation of the organ, with the anterior side of the epididymis tubule in close contact with the testis.

## Discussion

Using whole-mount multiplex immunolabeling, organ clearing, ultramicroscopy imaging, and 3D reconstruction with a highly adapted lymphatic-YFP transgenic reporter mouse model, the sensitivity and power of our study has allowed us to present for the first time an in-depth analysis of blood and lymphatic networks both in the adult and during postnatal epididymal ontogeny. With the combination of lymphatic markers used in this work, we have shown that a dense network of conventional lymphatics can be observed in the epididymis, with initial lymphatics present in the interstitial compartment and collecting lymphatics present at the septa. We have shown that the lymphatic network is denser on the anterior face of the epididymis (face adjacent to the testis) and progresses in the organ toward the posterior face. We found that the collecting lymphatics are mainly observed on the anterior surface and radiate inward toward the posterior surface following the septa. The initial lymphatics drain the interstitial compartment and join the collectors at the septa, resulting in an asymmetric distribution (anterior/posterior) of lymphatic vascularization. This centrifuge development of the lymphatic network is easily visible (*Figure 4A* and *Figure 2—video 2*). This tree structure is logical if we consider that lymphatic vascularization progresses during mouse development following the PP before reaching the *rete testis* and sprouting laterally, as had been suggested earlier (*Svingen et al., 2012*). Each epididymal septum has lymphatic drainage, but three septa appear to be more involved, including the S1/S2–S3, S6/S7, and S8–9/S10 junctions, with the latter having the largest lymphatic collector. We expected this finding because it corresponds to the position of the arteries and veins that irrigate the epididymis (*Suzuki, 1982*), and it is known that the large lymphatic collectors usually follow the path of the arteries (*Gruffaz, 1984*). Previous researchers have hypothesized that this strict epididymal segmentation is involved in the control of ascending pathogens, although they have not provided clues as to how this might be accomplished (*Stammler et al., 2015*; *Turner et al., 2003*). Supporting this hypothesis is the observation that large lymphatic collectors that are in direct and rapid connection with LNs and the blood supply that allows for the arrival of leukocytes are present at these sites. These three septa could then be considered important checkpoints (*Figure 8*). This view corroborates recent data showing that immune responses in autoimmune epididymitis are concentrated in the epididymal *cauda* and *corpus*, where we do see the major lymphatic collectors (*Wijayarathna et al., 2020*). Although we have arbitrarily chosen to focus our discussion on the specific involvement of the epididymal vascular compartment in inflammation and immune responses, it should be mentioned that it could also serve other processes such as tissue metabolism and the stem/progenitor cell niche microenvironment associated with different regions of this organ.

Looking at the postnatal development of the epididymal lymphatic network, our data show a progressive increase that follows the postnatal growth of the organ between 10 and 40 DPN. Beyond this time point, the lymphatic network progresses more in the S1/IS and S10 compartments. This

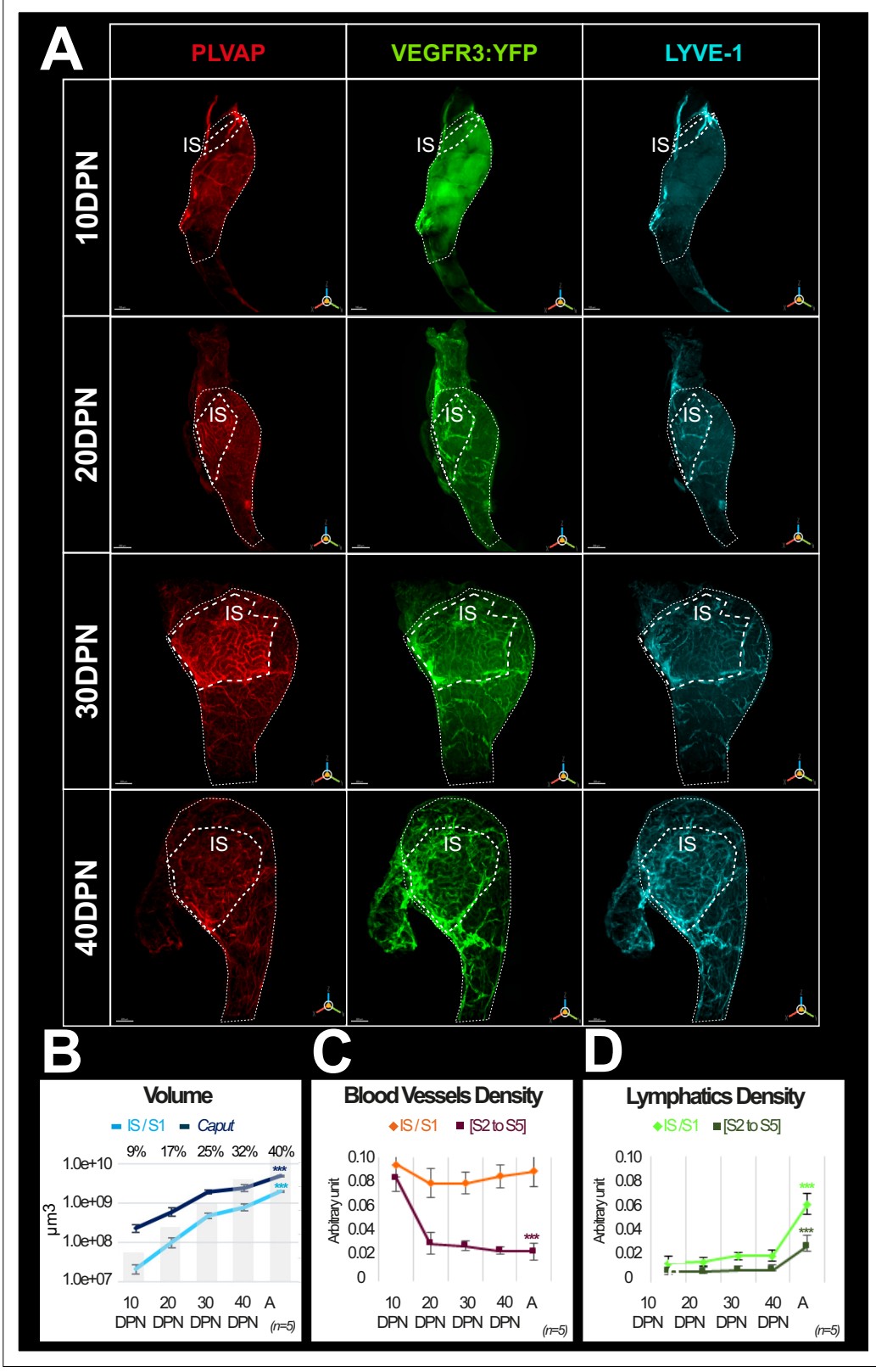

**Figure 4.** Evolution of the blood and lymphatic vasculature during postnatal epididymal development. Panel A shows representative three-dimensional (3D) images of blood and lymphatic networks at different postnatal stages during caput ontogenesis. Immunostaining was done with the blood vessel marker MECA32/PLVAP antibody (red) and the lymphatic marker LYVE1 antibody (cyan). The VEGFR3:YFP transgene is revealed by an anti-GFP antibody.

*Figure 4 continued on next page*

*Figure 4 continued*

Only LYVE1 and VEGFR3 are presented here for better resolution. Immunostaining was also performed with PDPN (not shown in the figure) but PDPN data can be seen in the data sources (figure 4 data sources 1,2 and *Figure 4— figure supplement 1* data source 1&2 at https://www.ebi.ac.uk/biostudies/bioimages/studies/S-BIAD618).The dotted line indicates the contours of the initial segment (IS = S1). DPN: days postnatal. The scale bar is 200 μm for 10 DPN, 300 μm for 20 and 30 DPN, and 400 μm for 40 DPN. Panel B shows the evolution of the volume (in log10) of the S1 segment (light blue curve) and the caput region (dark blue) during postnatal development. The superimposed histogram gives the proportion of volume occupied by the S1 relative to the caput region at different stages of postnatal development. Surface rendering of blood vessels and lymphatics was performed using IMARIS software. Blood and lymphatic vessel densities were calculated as described previously (*Figures 2 and 3*). Quantification was performed on five replicates for the different postnatal developmental stages. Panels C and D are graphs presenting densities (mean and standard error of the mean) of blood and lymphatic vasculature, respectively. The Kruskal–Wallis test with Dunn's posttest correction was used to determine statistical significance (***p<0.0001, **p<0.001, *p<0.01).

The online version of this article includes the following figure supplement(s) for figure 4:

**Figure supplement 1.** Evolution of the blood and lymphatic vascularization of the epididymal cauda during postnatal development.

argues for active lymphangiogenesis or/and lymphangio-vasculogenesis, the two modalities by which lymphatics develop, in the *caput* and *cauda* territories. Lymphangiogenesis corresponds to the extension of existing vessels, also known as 'sprouting,' while lymphangio-vasculogenesis involves the participation of precursor cells that create clusters later integrating already formed lymphatic vessels (*Gutierrez-Miranda and Yaniv, 2020*; *Jafree et al., 2021*). Lymphangio-vasculogenesis has been reported in the development of the mesenteric lymphatic vessels (*Stanczuk et al., 2015*) and for the heart and kidney lymphatic networks (*Jafree et al., 2019*; *Lioux et al., 2020*), two organs with a septal organization. The lymphatic neovascularization we observed during postnatal epididymal development is supported by the observation that lymphangiogenic ligands (VEGF-C and VEGF-D) and their corresponding receptor (VEGFR3) are concordantly present in the developing tissue. Active lymphangio-vasculogenesis is also supported by our observation that foci expressing LYVE1 can be detected between 10 and 20 DPN, and eventually extend and join the initial lymphatics (see *Figure 7B*).

*Suzuki, 1982*, reported that a testis-like blood network is present in most of the epididymis (segments 2–9) with intertubular vessels connected by perpendicular micro-vessels in a rope ladder organization. Distinctly, in the most proximal (IS/S1) and distal (S10) segments of the epididymis, we found micro-vessels encircling each tubule in a spider web organization with capillaries that penetrate the epithelial sublayer. Because the IS and S10 compartments of the epididymis have distinct embryonic origins and are functionally different (*Abe et al., 1984*; *Johnston et al., 2007*), it is difficult to explain this common vascular organization. We hypothesize that the common set-up of these two territories where capillaries penetrate the sub-epithelial layer could be associated with their immune function. The S10 compartment must monitor ascending pathogens and maintain self-tolerance to sperm antigens that accumulate in this storage part of the organ. Similarly, the S1 compartment is the final gatekeeper preventing ascending pathogens from reaching the immune-privileged seminiferous tubules, while at the same time it must be tolerant to sperm-specific antigens entering the epididymal tubule and considered non-self. Therefore, in both compartments, a spider web blood network can be expected to allow immune cells easy access to the epididymal tubule. This should be particularly true for the S1 segment because, independently of infectious situations, it is constantly solicited in the adult animal by the arrival of sperm antigens and by its function of reabsorption of Sertolian fluid requiring very permeable vessels. This hypervascularization of the S1 epididymal compartment is further supported by the fact that it has been shown to be PLVAP[pos], a marker of fenestrated vessels allowing high trans-endothelial transport (*Guo et al., 2016*). In other contexts, PLVAP has been shown to be required for fenestron biogenesis and organization, controlling fenestron permeability, angiogenesis, as well as leukocyte, antigen migration, and immomodulatory processes (*Rantakari et al., 2015*; *Amersfoort et al., 2022*). In addition, the presence of VEGFR2, VEGFR3, and VEGF-A in the microvascularization of the epididymis S1 compartment argues for active angiogenesis (*Bussolati et al., 2003*). This has been corroborated by the recent single-cell data (*Rinaldi et al., 2020*) showing that expression of angiogenic markers such as those mentioned above occurs in endothelial cells of

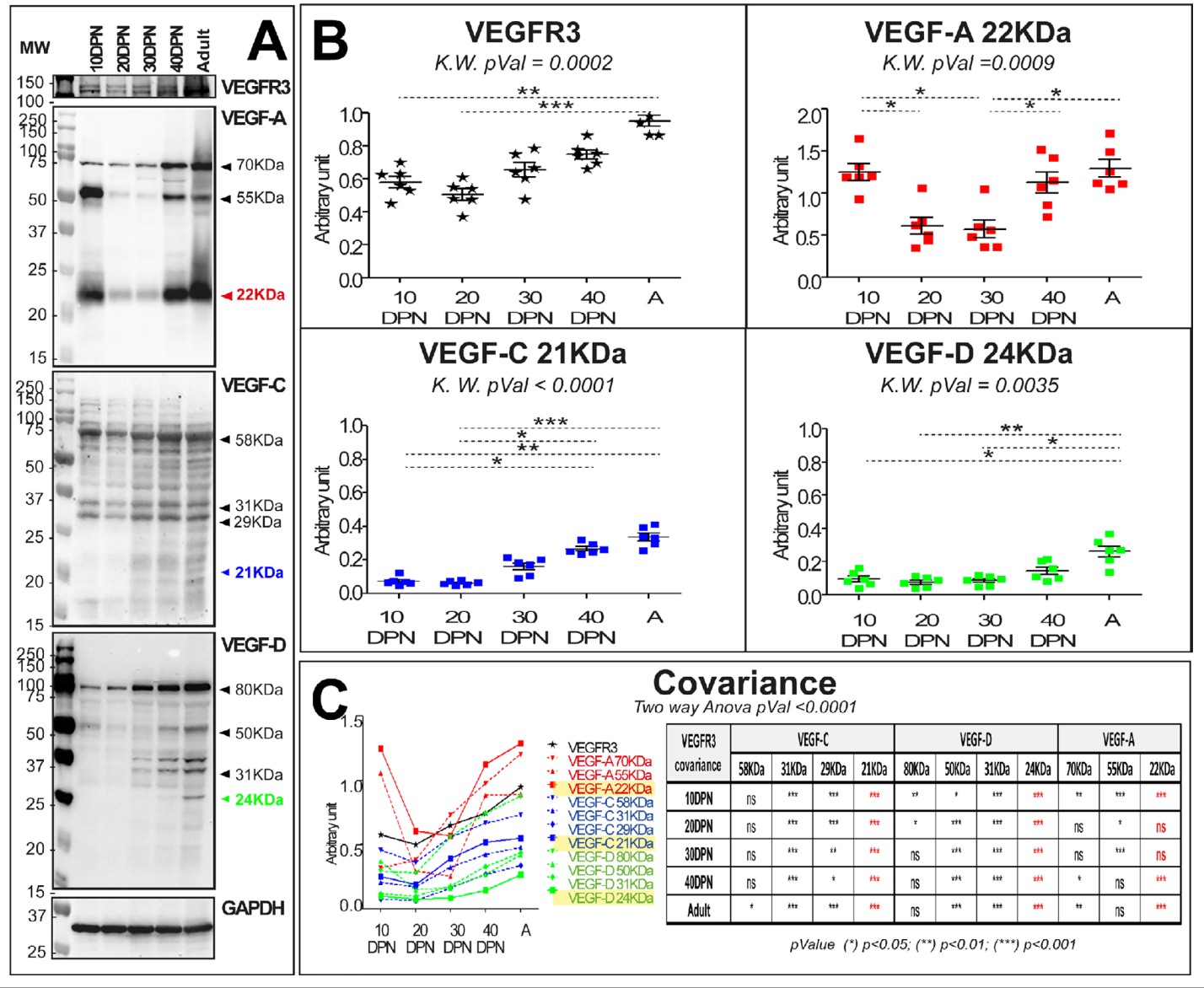

**Figure 5.** VEGF-A, VEGF-C, and VEGF-D levels vary during postnatal epididymal development. Panel A shows the expression profile obtained in total extracts of epididymal proteins at different stages of development. The profiles obtained are presented in the following order: VEGFR3, VEGF-A, VEGF-C, and VEGF-D. GAPDH was used for normalization. Panel B shows the quantification of epididymal proteins extracted from six mice. Quantification of VEGFR3 is shown in black, and VEGF-A, VEGF-C, and VEGF-D are shown in red, blue, and green, respectively. The Kruskal–Wallis test and Dunn's posttest correction were used to determine statistical significance (***p<0.0001, **p<0.001, *p<0.01). Panel C shows the comparison of different isoform profiles. Covariance is assessed by a two-way analysis of variance.

The online version of this article includes the following figure supplement(s) for figure 5:

**Figure supplement 1.** Quantification of all forms of the hemangiogenic and lymphangiogenic ligands VEGF-A, VEGF-C, and VEGF-D.

the caput mouse epididymis. While active angiogenesis during postnatal epididymal development makes sense because it accompanies postnatal growth of the organ, it is more surprising that angiogenesis is still active in the adult IS epididymal compartment. In adult tissues, angiogenesis is most commonly associated with vascular remodeling in tissue repair processes and tumor progression (*Komi et al., 2020*). To date, self-renewal of the epididymal epithelium is collectively assumed to be rather slow, a situation that should not be associated with high angiogenic activity. However, very recent data could modify this general assumption because it has been reported that the basal cells of the epididymis share common properties with adult stem cells (*Dufresne et al., 2022*). This could

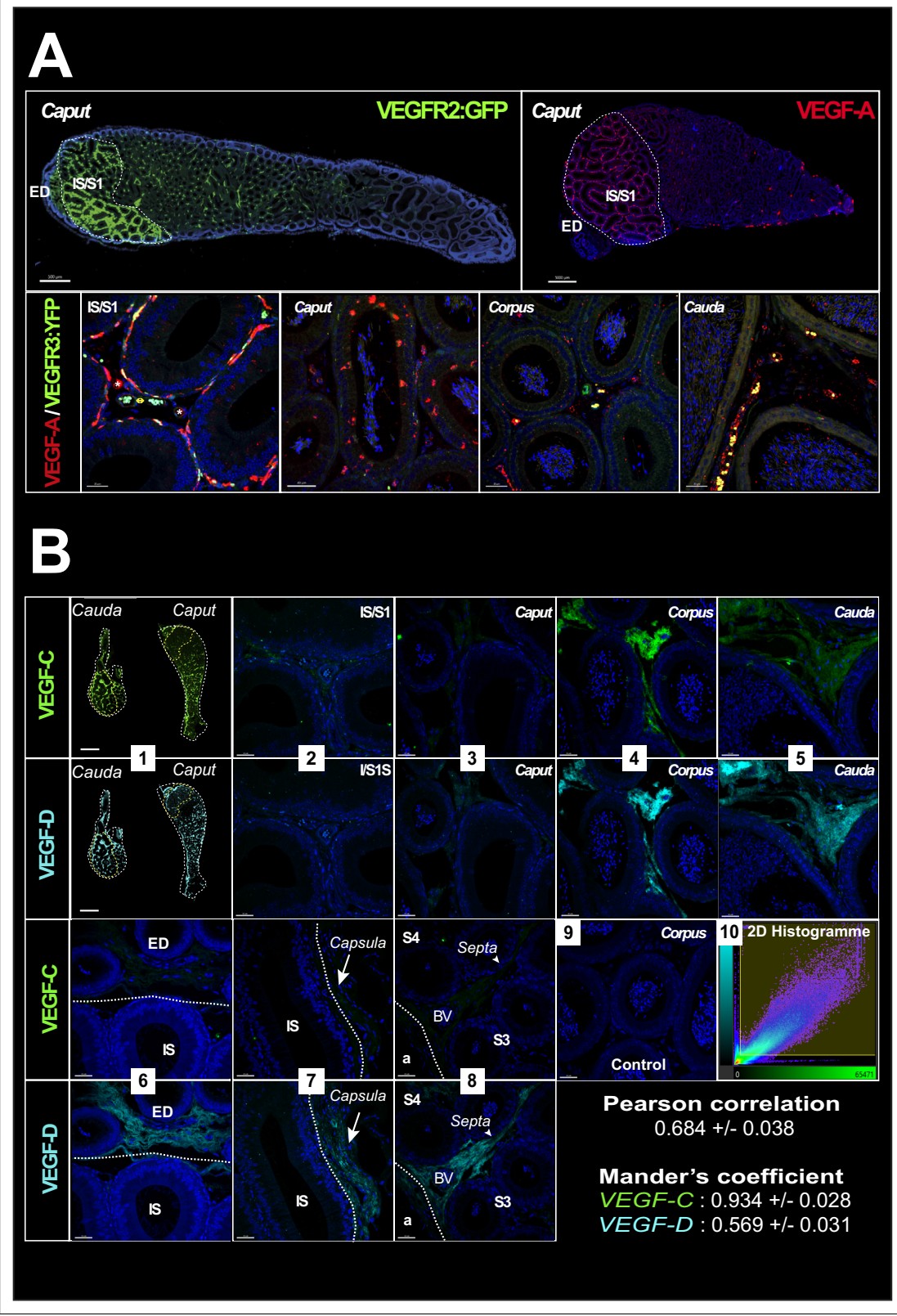

**Figure 6.** VEGF receptors and ligands in the mouse epididymis. Panel A (upper row) shows images (Zeiss Axio-Imager) of the expression of the VEGFR2:GFP transgene (green) (used with permission from Prof. A. Medvinsky) and its ligand VEGF-A (red) obtained from paraffin sections of adult epididymis. The lower row shows confocal views (SP8, Leica) of the same VEGF-A labeling. Panel B shows in (1) a mosaic view (AxioVison scanner) of an epididymis after immunolabeling with VEGF-C (green) and VEGF-D (cyan). Photographs 2–9 were taken with a confocal (SP8, Leica) at the level of

*Figure 6 continued on next page*

*Figure 6 continued*

the initial segment (IS)/S1 (2), caput (3), corpus (4), and cauda (5). Notable differences in expression of these two ligands are shown in photographs 6–8, which respectively concerns the efferent duct and the IS/S1 boundary, the capsule, and a septum. Photograph 9 is a negative control. Pearson correlation and Mander's co-occurrence were used to analyze the relation between the two ligands with the IMARIS co-localization module. The values shown represent means with standard error of the mean.

support the idea that the epithelium of the adult epididymis may be in a perpetual process of self-renewal that could be promoted by the quasi-inflammatory situation inherent to immune stimulation mediated by sperm antigens.

It is interesting to note that during postnatal ontogeny of the epididymis, PLVAP expression increases in the S1 territory but decreases in the other regions of the organ as early as 15 DPN.

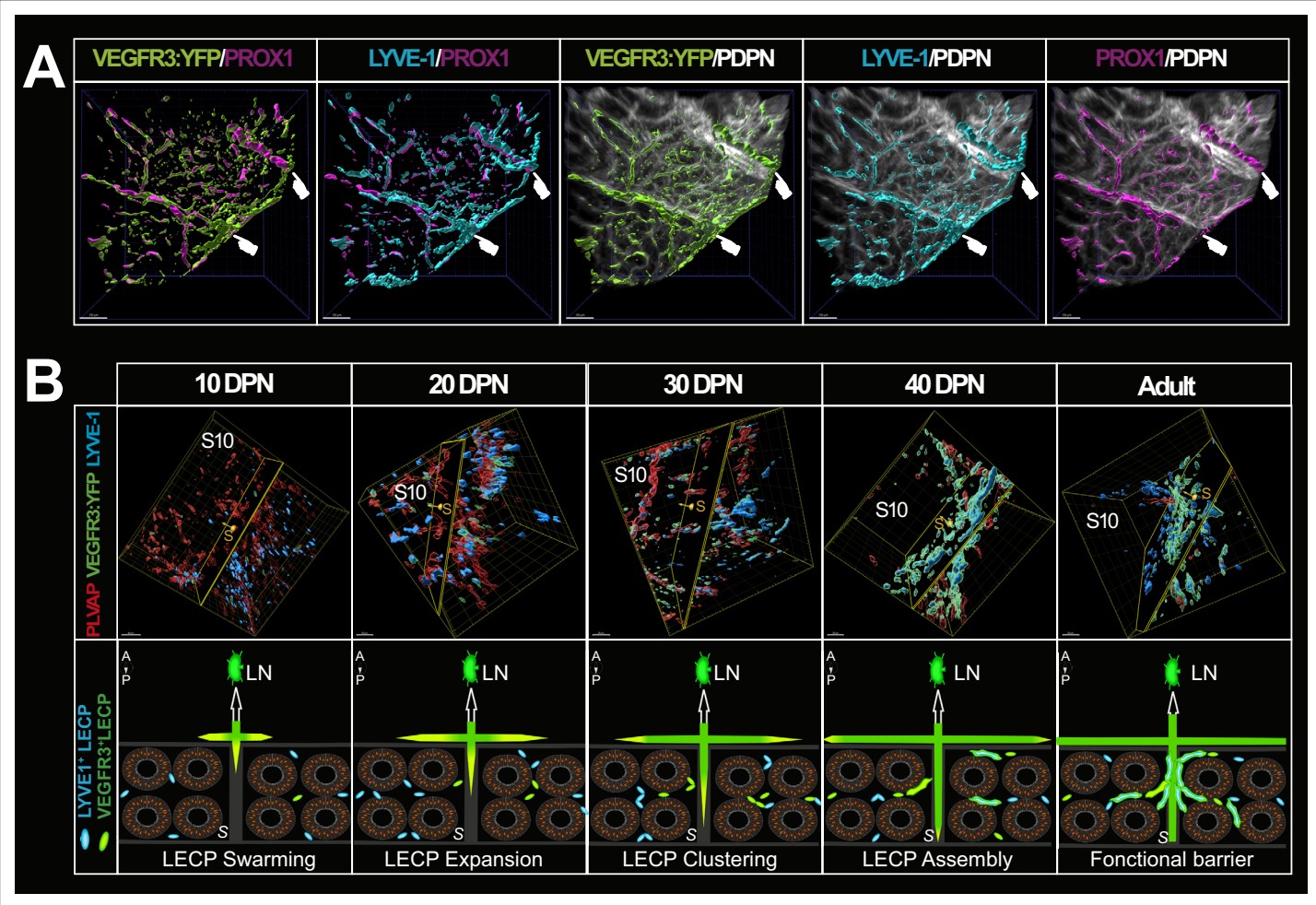

**Figure 7.** The septa between epididymal segments are intimately linked to lymphatic vasculature. Panel A shows the intimate connection between the septa and the lymphatic vasculature in a region of interest of the caput shown in *Figure 2*. Surface rendering of the lymphatics was performed as described previously using IMARIS software. Panel B (top panel) shows a representative three-dimensional (3D) view of the cauda region at the S8–S9/S10 septa during postnatal epididymal development after surface rendering. Images were obtained using the light-sheet ultramicroscope with a 20× objective lens. Putative LYVE1+ lymphatic precursors are in cyan and VEGFR3:YFP lymphatics are in green, whereas PLVAP+ blood vessels are in red. The clipping plane is positioned at the septum level and the yellow arrow-cursor is directed to the S10. The lower panel shows our interpretation of the events observed concomitantly with a progression via sprouting lymphangiogenesis (green to yellow arrows) of peripheral lymphatics that progress and radiate into the organ at the level of the septum, from the anterior side (adjacent to the testis) to the posterior side of the epididymis and via lymphangio-vasculogenesis. It develops in four steps: (1) swarming of precursors mainly LYVE+ (cyan) and VEGFR3+ (green); (2) a stage of LECP expansion; (3) grouping or clustering step; (4) then assembly and fusion with the lymphatics located at the septum from lymphangiogenesis (green to yellow arrow). For clarity, PLVAP+ blood vessels have not been represented here (A=anterior side adjacent to the testis, P=posterior side, S=septum, LN = lymph node). Scale bar = 30 µm.

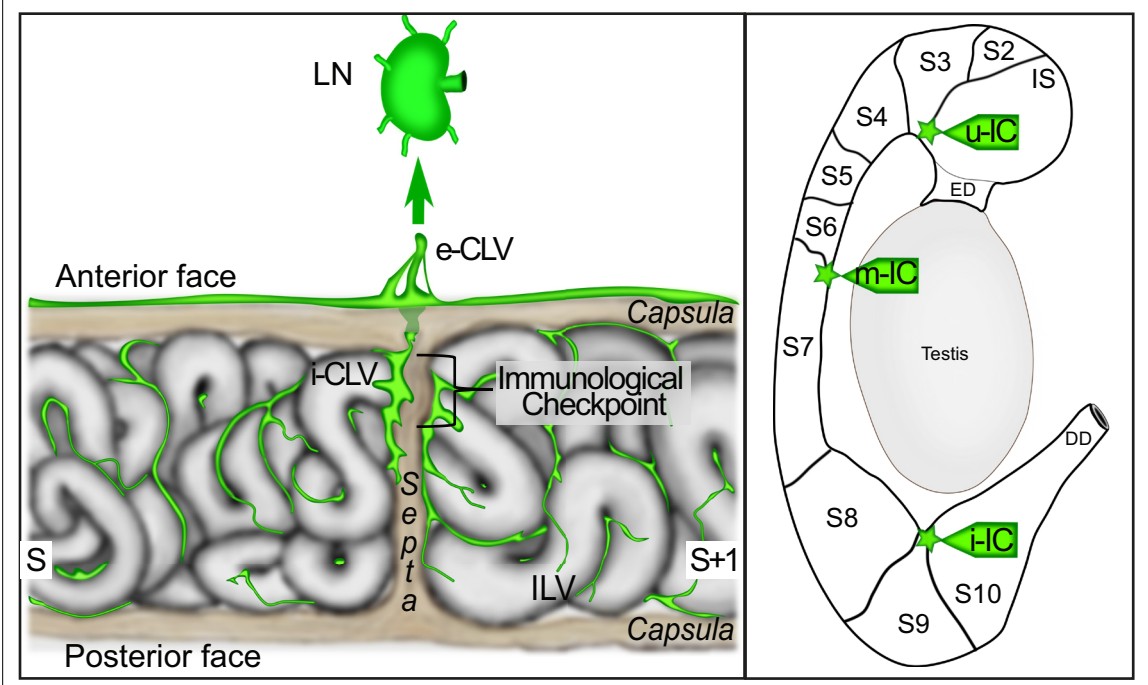

**Figure 8.** The proposed model of the lympho-septa of the epididymis. The left scheme presents our view of the lymphatic vasculature at the level of a 'checkpoint' septum. The large collectors enter at the septum into the epididymis and radiate within the adjacent segments at the interstitial level, where initial lymphatics are found. The septum where the tubule crosses from one segment to another one would be the most 'monitored' site and therefore the richest in lymphatics. The close association of lymphatics with the septa creates both a physical and immunogenic barrier to preserve the organ from ascending infections and thus limited uncontrolled progression of pathogens protecting the epididymis and ultimately the testis from orchitis deleterious to male fertility. e-CLV=external collector lymphatic vessel; i-CLV=internal collector lymphatic vessel; ILV = initial lymphatic vessel; LN = lymph node; S=segment.

Whether this is due to blood vessels loss or to PLVAP$^{pos}$ permeable fenestrons becoming PLVAP$^{neg}$, less permeable vessels will require further study. In the mouse model, it has been shown that the development of the S1 territory occurs between 15 and 20 DPN concomitantly with the arrival of testicular fluid (*Jegou et al., 1982*), which has been shown to stimulate the Raf/Mek/Erk pathway known to be activated by VEGFs (*Xu et al., 2016*). Therefore, it is possible to consider that VEGF-A could be one of the lumicrine factors triggering angiogenesis and differentiation of S1/IS. In agreement with this, it has been reported that mice overexpressing VEGF-A showed dilation of the *caput* and *corpus* epididymis, with enlarged and more permeable blood vessels (*Korpelainen et al., 1998*). More recently, *Pawlak and Caron, 2020*, in their review have suggested that such vessels could be hybrid lymphatics also recently called 'mosaic vessels' (*Lampejo et al., 2023*) expressing both blood and lymphatic markers. Such hybrid vessels have been found in distinct vascular beds, including high endothelial venules in secondary lymphoid organs, liver sinusoidal endothelial cells, Schlemm's canal endothelial cells of the eye, the ascending *vasa recta* (AVR) of the kidney inner medulla region, and the remodeled spiral arteries of the placenta decidua (*Gola et al., 2021*; *Kenig-Kozlovsky et al., 2018*; *Kim et al., 2017*; *Pawlak et al., 2019*; *Russell et al., 2019*; *Stamataki and Swadling, 2020*) where they perform specialized exchange functions. As shown in *Figure 9*, these structures share many markers, including those tested in the present work, suggesting that some of the assumed blood vessels found in the S1 epididymal territory might be hybrid vessels. Given the close mesonephric origin and similar fluid reabsorption function of the IS/S1 epididymal territory and the AVR, it is possible that the two structures share such hybrid vessels. Tie2 expression, a marker of all currently described hybrid lymphatics (*Kenig-Kozlovsky et al., 2018*, see *Figure 9*), supports this hypothesis as it localizes to the S1 peritubular microvascularization together with VEGFR3 and PVLAP (not shown).

In conclusion, based on our extensive study of the blood and lymphatic vascularization of the epididymis in mice, we propose that the epididymis presents two types of lymphatic vessels. On the one hand, there are peritubular PLVAP$^{pos}$ hybrid vessels, very permeable, particularly well represented

| Genes | Epididimys MRG Database (RNA) Low — High | IS vessels expression (IHC) | Ascending Vasa recta | Schlemm Canal | HEV | Remodeled Spiral Arteries | Liver Sinusoid |
|---|---|---|---|---|---|---|---|
| PLVAP | | Positive | Positive | Positive | Positive | nd | Positive |
| VEGFR2 | | Positive | Negative | Positive | Positive | nd | Positive |
| VEGFR3 | | Positive | Positive | Positive | +/- | Positive | Positive |
| PROX1 | | nd | Positive | Positive | Negative | Positive | Negative |
| LYVE-1 | | +/- | Negative | Negative | +/- | Positive | Positive |
| PDPN | | Negative | Negative | Negative | Negative | Negative | +/- |
| CD31 | | nd | Positive | Positive | Positive | Positive | +/- |
| CD34 | | nd | Positive | Positive | nd | Positive | Negative |
| Tie2 | | Positive (not shown) | Positive | Positive | Positive | Positive | Positive |
| Sox18 | | nd | nd | +/- | nd | nd | Positive |
| NRP2 | | nd | nd | nd | nd | Negative | Negative |
| EMCN | | nd | Positive | Positive | Positive | Negative | Positive |
| NRP1 | | nd | nd | Positive | Negative | Positive | Positive |
| CALCRL | | nd | nd | nd | nd | Positive / Positive | Positive / Positive |
| VEGFR1 | | Positive(*) | Positive | Positive | Positive | Positive | Negative |
| VE-Cad | | nd | nd | Positive | Positive | Positive | +/- |
| Foxc2 | | nd | nd | Positive | Positive | Positive | Positive |
| Ephb4 | | nd | nd | nd | nd | Positive | Positive |
| vWF | | nd | nd | Positive | nd | nd | nd |
| Itga9 | | nd | nd | Positive | nd | nd | Positive |
| KLF4 | | nd | nd | Positive | nd | nd | nd |
| Ephb2 | | nd | nd | Positive | nd | Positive | Positive |

**Figure 9.** Expression of hybrid lymphatic markers in the mouse epididymis. This table compares markers associated with hybrid lymphatic vessels described in the literature (for a review, see *Pawlak and Caron, 2020*) with (column 2) their expression at different segments of the epididymis (source: Mammalian Reproductive Genetic; https://www.mrgd.org). Multiple results for the same gene correspond to the use of different Oligosets in the microarrays used by the MRG. Column 3 summarizes our present results at the initial segment level. The asterisk refers to the publication by *Korpelainen et al., 1998*.

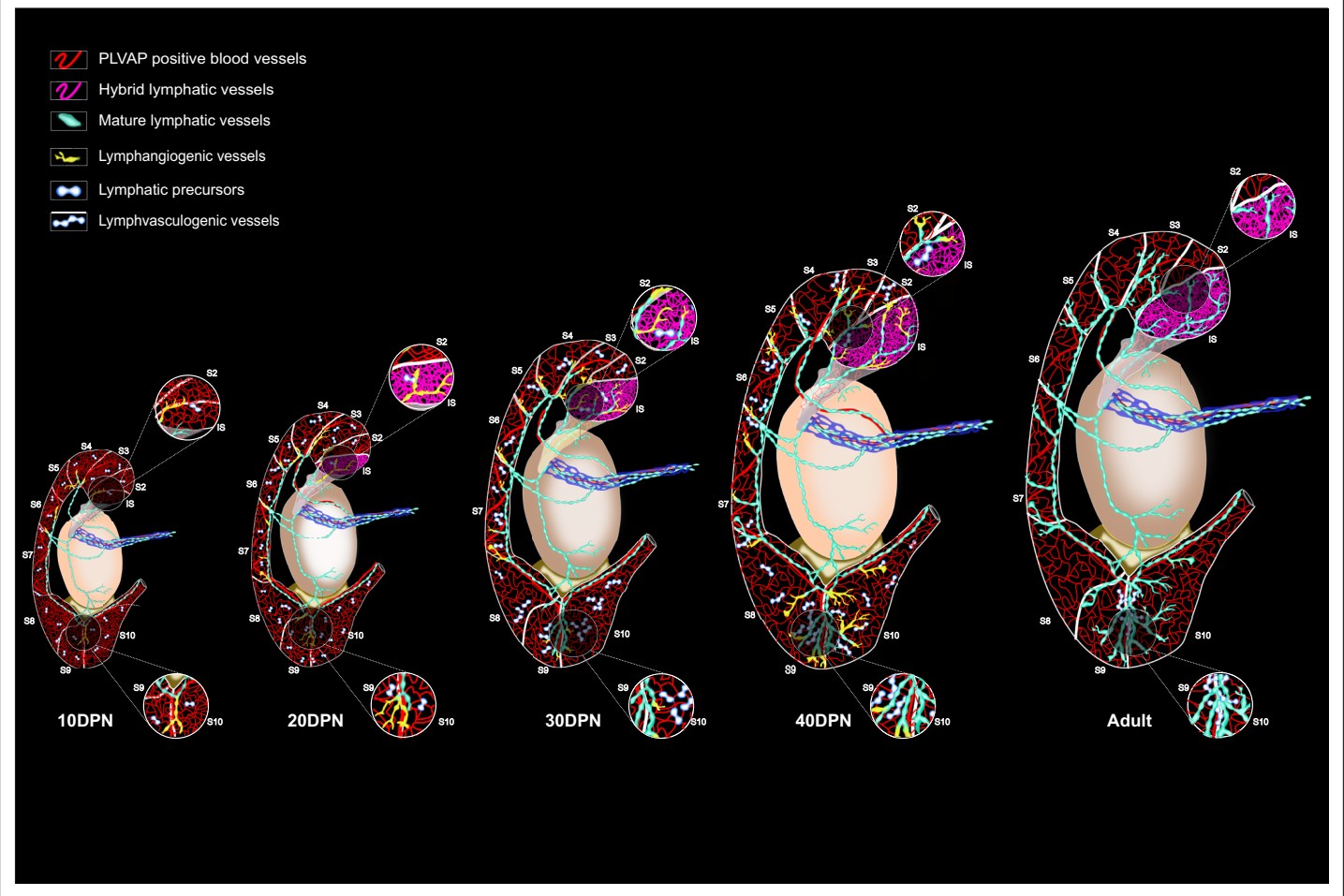

**Figure 10.** Schematic summary of the expansion of the conventional and hybrid lymphatic vasculature during postnatal development of the murine epididymis.

in the S1 compartment that, according to our hypothesis, participate both in the reabsorption function and in the immune surveillance of this territory. On the other hand, we have more conventional lymphatics mainly in the interstitial compartment. The various segments of the epididymis are drained by initial and collecting lymphatic vessels, the latter being closely associated with the septa and then connecting via the PP to the most proximal LN. Three checkpoints are represented by the IS/S2–S3, S6/S7, and S8–9/S10 septa. It is not yet clear to us that the lymphatic drainage of the S8–9/S10 compartment goes to the PP because it could also be drained by the lymphatic circuit monitoring the vas deferens. This detailed knowledge of the blood and lymphatic circuits of the epididymis highlights how open the epididymis is to the systemic compartment and how different its immune/inflammatory regulation is expected to be from that of the testis, where lymphatic collectors are restricted to the conjunctiva and never reach the interstitial compartment (*Hirai et al., 2012*; *Svingen et al., 2012*). A detailed schematic of epididymal vascular ontogeny in the mouse is proposed in *Figure 10*. If these observations in the mouse model are somehow translatable to the human epididymis, they could help to understand the complex inflammatory and immune contexts of posttesticular sperm maturation and their impact on male fertility.

## Materials and methods

### Mice

In this study, we have used 3- to 6-month-old male C57BL/6 mice (Janvier Labs, France); the male VEGFR3-YFP mice came from the J.L. Thomas laboratory (UPMC-Inserm, Paris, France). The

Vegfr3TYFP construct was generated from a BAC clone (RP23-65D23, Chori BACPAC resources) containing a 238kb geneomic fragment spanning the mouse Vegfr3 locus (*Calvo et al., 2011*). All animals were housed according to institutional guideline with a 12 hr photoperiod and food and water available ad libitum. Measures were taken to keep animal suffering to a minimum. The Auvergne Animal Experiment Ethics Committee (C2E2A) and the French Ministry approved all the following procedures for research (APAFIS authorization #12376-2017112913113962V3).

## Antibodies

The primary and secondary antibodies used in clearing methods are indicated in *Figure 11B*. Additional antibodies used are: rat anti-mouse VEGFR3 (#NB110-61018, Novus Biologicals, LLC, Bio-Techne SAS, France); goat anti-mouse VEGF-A (#AF-493-SP, Novus Biologicals, LLC, Bio-Techne SAS, France), rabbit anti-mouse VEGF-C (#NB110-61022, Novus Biologicals, LLC, Bio-Techne SAS, France), goat anti-VEGF-D (#AF469, R&D Systems, Bio-Techne SAS, France), and rabbit anti-GAPDH (SAB2108668, Sigma-Aldrich Chimie Sarl, France).

## Western blot analysis

Soluble epididymal proteins were prepared, as described previously (*Chorfa et al., 2021*), separated with sodium dodecyl sulfate–polyacrylamide gel electrophoresis (12% gel), and transferred to polyvinylidene fluoride membranes (Hybond ECL, Amersham Biosciences, Germany). Primary antibodies against the following protein were used: VEGFR3, VEGF-A, VEGF-C, VEGF-D, and GAPDH (served as a loading control). The appropriate HRP-conjugated secondary antibodies, goat anti-rabbit IgG or goat anti-mouse IgG (Abliance, France), were used to visualize the protein bands. Immunoreactive bands were detected by chemiluminescence (Clarity Western ECL Substrate Bio-Rad, France) using the ChemiDoc MP imaging system (Bio-Rad). Protein quantification was performed with ImageJ software. Protein amounts are expressed as relative values to the GAPDH.

## Whole-mount multiplex immunolabeling and clearing procedure

Epididymides were collected from mice after cervical dislocation and fixed in 4% paraformaldehyde for a period of time determined by their size (see *Figure 11*). Then, they were subjected to saturation/permeabilization for 1–5 days depending on their size in PBSGTS solution (1× saline buffer phosphate containing 2% gelatin, 0.5% Triton X-100, and saponin 1 µg/ml) on a rotary shaker (100 rpm) at room temperature. Incubation with primary and secondary antibodies was performed under rotation at 37°C in the same buffer according to the schedule described in *Figure 11A*. Concentrations of primary and secondary antibodies used are reported in *Figure 11B*. The immunolabeled epididymides were then embedded in a 1.5% agarose cube in 0.5× Tris–Acetate–EDTA buffer and cleared according to the 3DISCO method (*Belle et al., 2014*). The samples were stored in a dark place at room temperature until observation.

## Immunofluorescence

Paraffin-embedded 5 µm sections of epididymides were subjected to heat-induced antigen recovery and then permeabilized for 30 min at room temperature with PBS supplemented with 0.3% Tween 20 and saponin (1 mg/mL). The sections were incubated for 1 hr at room temperature in the same solution with 10% serum before overnight incubation with the primary antibody at 4°C. Their detection was then carried out with appropriate secondary antibodies conjugated to Alexa Fluor 488, Alexa Fluor 555, or Alexa Fluor 647 (Invitrogen, Thermo Fisher Scientific, France). Nuclei were stained with Hoechst solution (1 µg/µL). The sections were embedded in Mowiol 4-88 (Sigma-Aldrich Chimie Sarl, France) and stored at 4°C until observation.

## Image acquisition

### Macroscopic microscopy

Macroscopic observations of endogenous VEGFR3:YFP transgene expression of the epididymis and testis were performed using a Leica binocular magnifier (with a 1× objective).

### Confocal microscopy

Multiplex immunofluorescence acquisition was performed using a SP8 confocal laser scanning microscope equipped with a Plan Apo $\lambda$ 40× Oil objective (Leica, Germany). The following parameters

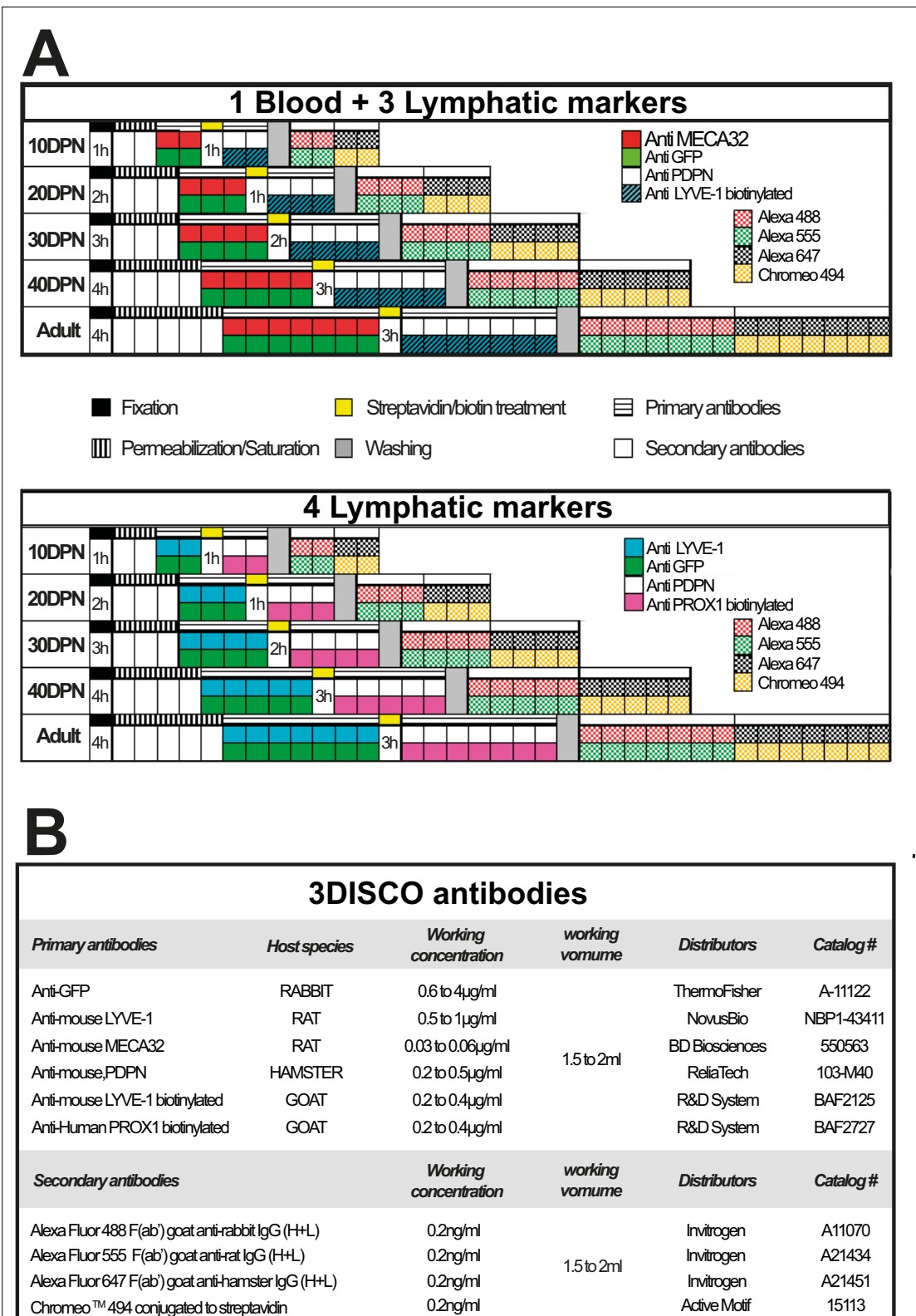

**Figure 11.** Multiplex labeling workflow used to visualize the blood and lymphatic vasculature of the mouse epididymis clarified by the 3DISCO method. Panel A presents the multiplex labeling schedule for blood and lymphatic immunodetection. Each square corresponds to 24 hr except for the fixation and streptavidin/biotin treatment, where the time is indicated. Panel B provides detailed information regarding the various antibodies used in the course of the study.

were used: pinhole size of 1 airy unit (AU); Z-step: 0.5 μm. Four acquisition sequences were executed: (1) for Alexa 647 detection (638 nm laser 659–698 nm window, using a photomultiplier tubes (PMT) detector at 700 V gain); (2) for Alexa 488 detection (488 nm laser; 493–541 nm window, using a PMT detector at 565 V gain) and for Chromeo 494 detection (488 nm laser; 660–700 nm window, using a PMT detector with a gain of 800 V); (3) for Alexa 555 detection (laser 552 nm; 560–570 nm window, using a PMT detector with a gain of 700 V); and (4) for Hoechst detection (laser 405 nm; 423–518 nm window, using a Hybrid detector with 10% gain).

## Light-sheet ultramicroscopy

3D imaging was achieved with an ultramicroscope (LaVision BioTec Miltenyi, Germany) using ImspectorPro software. The images were obtained with either an MI PLAN 2×/NA 0.5, an MI PLAN 4×/0.35 objectives (MVPLAPO, Olympus), or a 20×/0.95 objective (Leica). Each sample was placed in a 100% quartz imaging reservoir filled with dibenzyl ether and illuminated from the side by the laser sheet. Images were acquired with a Andor Neo SCMOS CCD camera (2160×2,160 pixels, LaVision BioTec). All 3D acquisitions were performed with a step size between each image was fixed at 1 μm (1.6 μm/pixel). Lasers at 488, 561, and 635 nm were used to obtain images using two light sources for the *caput*, *corpus*, or *cauda* epididymis.

## Image processing

Ultramicroscopy datasets were uploaded to IMARIS version 9.7 (Bitplane, Oxford Instruments, England). The stacks were converted to IMARIS files (.ims) and 3D visualizations of z-stack images were generated using the volume rendering function. The vessels were transformed into surface rendering and the volume calculated automatically by IMARIS. Vessel densities were estimated as the ratio of the vessel volume to the total volume of the region. The videos were made IMARIS without deconvolution of the image.

## Statistical analysis

All experiments were repeated at least three times and representative images are shown. All statistical analyses were performed with Prism software (GraphPad, USA). The Mann–Whitney test was used to compare two groups. The Kruskal–Wallis test followed by Dunn's posttest was used to compare more than two groups. A two-way analysis of variance test with a Bonferroni posttest was used to analyze the covariance of ligands (VEGF-A, VEGF-C, and VEGF-D) with the VEGFR3 receptor. The co-localization of the ligands VEGF-C and VEGF-D was evaluated by Pearson correlation and Mander's overlap coefficient with IMARIS co-localization modules (Bitplane). The values for co-localization analysis represent the mean ± standard error of the mean.

## Acknowledgements

The authors would like to thank the CLIC confocal microscopy platform (GReD Institute) for their expert assistance during this study.

## Additional information

### Funding
No external funding was received for this work.

### Author contributions
Christelle Damon-Soubeyrand, Areski Chorfa, Chantal Goubely, Nelly Pirot, Laurence Piboin-Fragner, Caroline Vachias, Stephanie Bravard, Ayhan Kocer, Investigation; Antonino Bongiovanni, Software, Investigation; Luc Pardanaud, Rachel Guiton, Jean-Leon Thomas, Fabrice Saez, Validation; Meryem Tardivel, Validation, Investigation; Joël R Drevet, Conceptualization, Supervision, Validation, Writing – original draft, Writing – review and editing; Joelle Henry-Berger, Data curation, Formal analysis, Investigation, Writing – original draft, Writing – review and editing

## Author ORCIDs

Antonino Bongiovanni http://orcid.org/0000-0001-9876-2940
Joël R Drevet http://orcid.org/0000-0003-3077-6558
Joelle Henry-Berger http://orcid.org/0000-0002-5107-8663

## Ethics

The Auvergne Animal Experiment Ethics Committee (C2E2A) and the French Ministry approved all the following procedures for research (APAFIS authorization # 12376-2017112913113962 V3).

## Decision letter and Author response

Decision letter https://doi.org/10.7554/eLife.82748.sa1
Author response https://doi.org/10.7554/eLife.82748.sa2

## Additional files

### Supplementary files

• MDAR checklist

• Supplementary file 1. Describes all the data source legends in (*.ims) (Imaris viewer) and (*.czi) (Zeiss AxioScan viewer) format that can be found in the BioImage Archive Biostudies EBI repository.

• Supplementary file 2. Is an Excel file listing all the data sources for the figures provided.

### Data availability

Data sources are available at https://www.ebi.ac.uk/biostudies/bioimages/studies/S-BIAD618. Data source legends are provided as Supplementary files 1 & 2. These files list the source data files available via the BioImage Archive repository BioStudies accession number S-BIAD618.

The following dataset was generated:

| Author(s) | Year | Dataset title | Dataset URL | Database and Identifier |
|---|---|---|---|---|
| Bongiovanni A, Damon-Soubeyrand C, Chorfa A, Goubely C, Pirot N, Pardanaud L, Pibouin-Fragner L, Vachias C, Bravard S, Guiton R, Thomas J, Saez F, Kocer A, Tardivel M, Drevet JR, Henry-Berger J | 2023 | Three-dimensional imaging of vascular development in the mouse epididymis : a prerequisite to better understand the post-testicular immune context of spermatozoa | https://www.ebi.ac.uk/biostudies/bioimages/studies/S-BIAD618 | EBI BioImage Archive, SBIAD618 |

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
