## [Editor Report]

This fundamental work substantially advances our understanding of vessel development in mouse epididymis. The evidence supporting the conclusions is compelling, with rigorous state-of-the-art microscopy involving high-resolution Three-dimensional imaging. The work will be of broad interest to cell biologists and male reproductive biologists.

---

## [Decision Letter]

**Decision letter after peer review:**

Thank you for submitting your article "Three-dimensional imaging of vascular development in the mouse epididymis: a prerequisite to better understand the post-testicular immune context of spermatozoa" for consideration by *eLife*. Your article has been reviewed by 2 peer reviewers, and the evaluation has been overseen by a Reviewing Editor and Mone Zaidi as the Senior Editor. The reviewers have opted to remain anonymous.

Essential revisions:

1) Revise Figure 2, 3, 4, and 7. Reorganization of the panel, higher magnification inserts for clarity.

2) Introduction should include some basics about lymphatic and blood vessels in the context of the epididymis.

3) Examine fetal stages to confirm when exactly vasculature is established.

4) VEGF expression data as presented is not convincing. Revise these data.

5) Revise the Discussion with a focus on major findings.

*Reviewer #1 (Recommendations for the authors):*

I find it misleading that the authors frame this study as a way to understand the immune context of the spermatozoa in the epididymis since there is no data on immune cells or immune responses. I suggest reducing the emphasis on the immune context by revising the title, abstract, and introduction, to reflect the true goal and context of the study, which is to update our current understanding of the circulatory networks of the epididymis by providing a detailed 3D description of vasculature and lymphatics. The fact that this description is an important piece of the immune and inflammatory responses could be reserved for the second half of the introduction and/or the discussion.

Graphical abstract: the differences between the different vessels and between the different stages of development are difficult to appreciate. Maybe the authors could just represent 2-3 stages and have blow-ups of the important differences?

It would be helpful for non-expert readers if the manuscript had a short introduction to the architecture of the lymphatic system and of the epididymis. For example, if the authors could define differences between collectors, nodes, and vessels, and explain the segmentation of the epididymis and the notion of septa, the manuscript would be more accessible to a broader audience.

Figure 2A: the diagram should be in the same orientation as the microscopy images.

For figures 2B and C, the surface renderings are difficult to appreciate, it would be helpful to provide blowups on the S1 instead of coloring it white in the low mag image. This will allow the reader to see better details of the density and complexity of the networks.

The titles for Figures 2 and 3 are almost identical, what more information is presented in figure 3? This should be reflected in the figure title.

In Figures 2, 3, and 4 it would help to add merged images of all the stains with high mage insets to better appreciate the amount of overlap and comparison between the markers, similar to what is shown in figure 3D.

I was not able to interpret figure 7A. The slicers that are meant to represent the septa are barely visible, and should all be presented in the same orientation/position to better appreciate the differences between the stages. The mix of native data and surface renderings makes it really difficult to view the vessels in these images. The diagram in 7A (second row) does not look enough like the images to help make sense of what is shown and what the authors hypothesize. Vessel types should be colored differently, as it is currently very difficult to tell the difference between LYVE1+ LECP and VEGFR3+ LEC

The discussion should recapitulate the major findings of the study and would benefit from being much more concise.

---

## [Author Response]

Essential revisions:1) Revise Figure 2, 3, 4, and 7. Reorganization of the panel, higher magnification inserts for clarity.

As recommended, we have rearranged most of the figures, with higher magnification where possible, for better clarity. In addition, because the figures in the revised manuscript are now presented individually without their legends, another magnification step took place accordingly.

2) Introduction should include some basics about lymphatic and blood vessels in the context of the epididymis.

At the request of the editors/reviewers, we now provide more information on epididymal blood and lymphatic vessels in the Introduction section (pages 3 and 4, lines 98 to 111). These statements were already in the original manuscript but were instead presented in the Discussion section. We have therefore brought them forward, which will also serve one of the editorial comments which was to shorten the Discussion section.

3) Examine fetal stages to confirm when exactly vasculature is established.

This point raised by Reviewer 2 (item 3) will be discussed below.

4) VEGF expression data as presented is not convincing. Revise these data.

This point raised by Reviewer 2 (item 4) will be discussed below.

5) Revise the Discussion with a focus on major findings.

By deleting an entire section in the first paragraph of our original Discussion, we believe we have condensed it and improved the strict presentation of our data. For the remainder of our Discussion, we do not feel that it needs to be greatly modified because we have tried to relate our data to the existing literature, as the situation of the epididymis having to deal with effective immune surveillance associated with strong immune tolerance is not limited to this territory. We have been very careful not to be too speculative and not to overuse our data. The discussion may seem a bit long, at least to one of the two reviewers, but the points that have been made cover the most recent bibliography, which we think should be the case in any discussion worthy of the name. In a final paragraph, we open the discussion to the hot topic of "hybrid lymphatic vessels" and develop the arguments that support the parallel we are making.

Reviewer #1 (Recommendations for the authors):I find it misleading that the authors frame this study as a way to understand the immune context of the spermatozoa in the epididymis since there is no data on immune cells or immune responses. I suggest reducing the emphasis on the immune context by revising the title, abstract, and introduction, to reflect the true goal and context of the study, which is to update our current understanding of the circulatory networks of the epididymis by providing a detailed 3D description of vasculature and lymphatics. The fact that this description is an important piece of the immune and inflammatory responses could be reserved for the second half of the introduction and/or the discussion.

With all due respect, we do not fully agree with the reviewer's comment. We believe that to understand the reasoning behind this particular study, one must fully embrace the systemic context of the male reproductive system and the outstanding issues. It is this overall picture that we believe should allow the reader to see how the work we present can help understand larger questions that remain not totally clear to date. Having done this in the first introductory paragraph, we then focus on the specific topic of the study being conducted. Therefore, if the editor does not object to this choice, we prefer that the abstract and the first paragraph of the introduction section remain unchanged. In partial response to the reviewer's request, and because it is true that we do not strictly present data on immune cells/molecules/responses (which we have done previously: Voisin et al., 2018; Voisin et al., 2020) in this manuscript, we have restricted the title of the manuscript to the epididymal vascular network.

Graphical abstract: the differences between the different vessels and between the different stages of development are difficult to appreciate. Maybe the authors could just represent 2-3 stages and have blow-ups of the important differences?

By taking advantage of the increased format available for the graphical abstract when submitting revised documents, we have made better use of space without removing some of the developmental steps where we think something clearly changes. We also now provide enlarged inserts for each step that we hope will improve the perception of this graphical abstract.

It would be helpful for non-expert readers if the manuscript had a short introduction to the architecture of the lymphatic system and of the epididymis. For example, if the authors could define differences between collectors, nodes, and vessels, and explain the segmentation of the epididymis and the notion of septa, the manuscript would be more accessible to a broader audience.

As already mentioned above in response to the reviewer's recommendations, we now provide in the Introduction section more information on epididymal blood and lymphatic vessels (pages 3 and 4, lines 98-111). Following the reviewer's suggestion, we have also provided more information regarding the variety of existing lymphatic structures (pages 4, lines 115-117) and referring to a very recent and detailed review on this topic (Lampejo et al., 2023). In addition, similarly following the reviewer's suggestion, we also briefly but hopefully more clearly describe the segmented histology of the epididymis (page 3, lines 72 to 77), again referring the reader to the literature if more detail is needed (Tuner et al., 2003; Johnston et al., 2005).

Figure 2A: the diagram should be in the same orientation as the microscopy images.

Corrected as suggested by the reviewer.

For figures 2B and C, the surface renderings are difficult to appreciate, it would be helpful to provide blowups on the S1 instead of coloring it white in the low mag image. This will allow the reader to see better details of the density and complexity of the networks.

Following the reviewer's recommendation, we have enlarged Figures 2B and 2C by isolating them in a separate figure (new Figure 3 in the revised version). In addition, the corresponding surface rendering graphs have been merged, hopefully making them more readable.

The titles for Figures 2 and 3 are almost identical, what more information is presented in figure 3? This should be reflected in the figure title.

The new design of figures 2 and 3 eliminated the old redundancy. The titles of the figures have been modified accordingly.

In Figures 2, 3, and 4 it would help to add merged images of all the stains with high mage insets to better appreciate the amount of overlap and comparison between the markers, similar to what is shown in figure 3D.

In our opinion, the videos we provide as supplementary data meet this demand better than any non-dynamic image, allowing an accurate overlay of the different networks.

I was not able to interpret figure 7A. The slicers that are meant to represent the septa are barely visible, and should all be presented in the same orientation/position to better appreciate the differences between the stages. The mix of native data and surface renderings makes it really difficult to view the vessels in these images. The diagram in 7A (second row) does not look enough like the images to help make sense of what is shown and what the authors hypothesize. Vessel types should be colored differently, as it is currently very difficult to tell the difference between LYVE1+ LECP and VEGFR3+ LEC

The changes we have made to the figure should clarify this point. However, it is difficult to orient all slicers in the same way because we will lose information for some stages. With the higher magnification now proposed, the figure should be more readable. Similarly, the schematic has been modified to enlarge the LECPs allowing better distinction from each other. The color choices reproduce the 3D staining, with VEGFR3 in green and Lyve1 in cyan.

The discussion should recapitulate the major findings of the study and would benefit from being much more concise.

As suggested by the reviewer, we have tried to make the discussion more concise and better present our results at the beginning of this section.